# History of Glial Cell Line-Derived Neurotrophic Factor (GDNF) and Its Use for Spinal Cord Injury Repair

**DOI:** 10.3390/brainsci8060109

**Published:** 2018-06-13

**Authors:** Melissa J. Walker, Xiao-Ming Xu

**Affiliations:** 1Medical Neuroscience Graduate Program, Indiana University School of Medicine, Indianapolis, IN 47450, USA; melissaw@ucla.edu; 2Spinal Cord and Brain Injury Research, Indiana University School of Medicine, Indianapolis, IN 47450, USA; 3Stark Neurosciences Research Institute, Indiana University School of Medicine, Indianapolis, IN 47450, USA; 4Department of Neurological Surgery, Indiana University School of Medicine, Indianapolis, IN 47450, USA

**Keywords:** spinal cord injury, glial cell line-derived neurotrophic factor (GDNF), GFRα-1, cRET, Schwann cells, astrogliosis, neuroprotection, axonal regeneration, combinational therapies, neurotrauma

## Abstract

Following an initial mechanical insult, traumatic spinal cord injury (SCI) induces a secondary wave of injury, resulting in a toxic lesion environment inhibitory to axonal regeneration. This review focuses on the glial cell line-derived neurotrophic factor (GDNF) and its application, in combination with other factors and cell transplantations, for repairing the injured spinal cord. As studies of recent decades strongly suggest that combinational treatment approaches hold the greatest therapeutic potential for the central nervous system (CNS) trauma, future directions of combinational therapies will also be discussed.

## 1. SCI Background and Need for Therapies

Spinal cord injury (SCI) is a devastating chronic condition, for which no effective treatments currently exist. The primary cause of SCI cases worldwide is motor vehicle accidents, followed by falls and sports injuries [1]. The potential long-term effects of chronic pain, inflammation, and devastating disabilities that SCI patients endure are compounded by the extensive lifetime costs of care. Approximately 1–5 million United States dollars (USD) is spent over the lifetime of an SCI patient, depending on the patient’s age and level of injury (NSCISC-National Spinal Cord Injury Statistical Center, 2018). The national cost of current and future healthcare for patients suffering from SCI in the United States is estimated to be more than 400 billion USD.

The initial SCI mechanical trauma disrupts local vasculature and leads to a breakdown of the blood–spinal cord barrier [2,3,4]. This is followed by a secondary wave of injury [5], comprised of hemorrhage, ischemia [6], excitotoxicity, edema, neuronal apoptosis, loss of gray and white matter tissue [6], axonal dieback, chronic inflammation [7], and the formation of a dense astrocytic glial scar surrounding the lesion. During the acute phase after SCI, the astrogliosis is presumed to be a positive regulator in limiting the spread of excitotoxic molecules, thus limiting the lesion area. For decades, the astrocytic glial scar has been considered inhibitory in chronic phases after SCI. However, recent literature supports beneficial axon regeneration in response to the astrocytic scar formation [8]. Glial cell line-derived neurotrophic factor (GDNF) has been shown to positively modulate astrogliosis [9,10,11], in addition to its known neuroprotective effects, thus making astrocytes a potential therapeutic target in experimental SCI.

## 2. Discovery of GDNF Family Ligands and Receptors

The GDNF subfamily of neurotrophic ligands consists of GDNF, neurturin (NRTN), artemin (ARTN), and persephin (PSPN), which bind to the glycosylphosphatidylinositol-anchored GFRα receptors 1–4, respectively [12]. The molecular structures of the GDNF family ligands and receptors are nicely detailed by the authors of [13], as well as in Figure 1. While ARTN [14,15], NRTN [11,16,17], and PSPN [18,19] have all been shown to be neuroprotective, this mini review focuses specifically on GDNF and its applications for the treatment of SCI.

GDNF was first identified as a neurotrophic factor released from glial cells by Engele et al. [20] and Lin et al. [21], in its promotion of the survival of dopaminergic neurons. The GFRα-1 receptor was first reported in Cell in 1996 [22], following its isolation, cloning, and characterization from rat retinal cells; a study which also detailed the interaction between GDNF, GFRα-1, and the cRET receptor. Interestingly, the following week a Nature publication [23] revealed concurrent work with similar findings on a cloned and characterized GFRα-1, as well as the GDNF, GFRα-1, and cRET multi-subunit receptor complex.

## 3. Localization of GDNF and its Receptors

Expression patterns of GDNF, GFRα-1, and cRET indicate that the three are not mutually exclusive for GDNF’s trophic actions, as GFRα-1 is expressed in regions lacking cRET, and cRET has expression in regions lacking GFRα-1 expression, well characterized by the authors of [24]. In 1996, Trupp et al. [25] identified GDNF’s activation of the cRET proto-oncogene, resulting in neuronal survival, while Jing et al. [22] identified GFRα-1 as mediating the interaction between GDNF and cRET. In 2001, Nicole et al. [26] demonstrated the expression of GDNF mRNA and protein, as well as GFRα-1 and cRET on both neurons and astrocytes. Heparan sulphate, a key glycosaminoglycan, was identified as crucial for the phosphorylation of the c-Ret co-receptor, and thus, also necessary for GDNF signaling through its GFRα-1 receptor [27].

Satake et al. [28] showed a dramatic upregulation of GDNF mRNA expression within 3 h post SCI that was maintained for approximately 2–4 weeks following injury. Additionally, changes in GDNF’s expression pattern following central nervous system (CNS) injury are nicely illustrated by Trupp et al. [12,25] and Donnelly and Popovich [29]. These studies provide potential time windows following SCI during which GDNF might be most effective. GDNF’s targets in the CNS and peripheral nervous system (PNS), as well as the administration of GDNF gene therapy for motoneuron protection were highlighted in a review by Bohn [30].

## 4. GDNF Promotes Cell Survival and Growth

One of the earliest studies to report GDNF-induced reduction of astrogliosis was a study by Trok et al. [31], in which spinal cord explants were allotransplanted into Sprague-Dawley anterior eye chambers. GDNF was shown to promote graft survival and growth, in addition to the reduced Glial fibrillary acidic protein (GFAP) immunoreactivity. Klöcker et al. [32] identified a new subpopulation of neurons responsive to GDNF in a study showing significantly reduced cell death of axotomized retinal ganglion cells in response to GDNF treatment. The upregulation of GDNF in the distal portion of peripheral injured nerves was assessed and quantified, along with the localization of its cRET receptor, as reported by Bär et al. [33]. Similarly, Höke et al. [34] showed upregulation of the GFRα1 receptor on the distal segment of the sciatic nerve following injury; this upregulation and the upregulation of GDNF by Schwann cells was maintained for approximately six months following injury. The GFRα1 receptor was localized to peripheral Schwann cells in a study by Hase et al. [35], showing another putative target of GDNF for the repair of the injured nervous system, particularly given that Schwann cells are the only cell type currently FDA approved for clinical trials of SCI in the United States [36]. Arce et al. [37] reported a 75% inhibition of neuron survival after exposure to Schwann cell cultured media containing a blocking antibody against GDNF, thus demonstrating the importance of GDNF for the Schwann cell-mediated neuroprotection. Paratcha et al. [38] highlighted the recruitment of cRET to neuronal cell membrane lipid rafts, in response to soluble GFRα1. Rind et al. [39] showed anterograde transport of GDNF in dorsal root ganglia (DRG) and motor neurons, both with undetectable levels of GDNF mRNA in their current state. The radiolabeled GDNF in this study was provided to the DRGs and motor neurons by Schwann cells and oligodendrocytes respectively, thus indicating another potential cell source of GDNF within the nervous system. In 2004, a novel in vivo study was published, showing for the first time the endogenous release of GDNF from astrocytes, which was neuroprotective to neighboring neuronal populations, in utero during development [40]. The production of GDNF from astrocytes, oligodendrocytes, and Schwann cells has made these cells important potential points of intervention for SCI therapies.

## 5. Molecular Signaling of GDNF for Promotion of Cell Survival

In addition to its neuroprotective effects [38,41,42], GDNF has also been shown to: (1) attenuate astrocyte cell death via reduced activation of caspase-3 [43], as well as through caspase-3/Akt independent mechanisms [44]; (2) minimize activation of microglia and production of nitric oxide [43,45]; and (3) promote the survival [46] and proliferation [40,47] of Schwann cells. GDNF activates rat primary cortical microglial cells through GFRα-1 and cRET receptors, with downstream signaling through the MAPK pathway, as illustrated in a study by Honda et al. [47]. Moreover, a pro-inflammatory response, resulting in increased levels of IL-1β, likely led to the GDNF neuroprotection observed in a lipopolysaccharide (LPS)-induced nigral degeneration model of Parkinson’s disease [48]. These studies demonstrate microglia as another putative therapeutic target for GDNF in CNS injury and disease.

Soler et al. [49] characterized the downstream signaling of GDNF in motoneurons, which includes activation of both the PI3K and ERK-MAPK pathways. Further investigation revealed that the neuroprotective effects of GDNF signaled through the PI3K pathway [49]. In 2001, Nicole et al. [50] described a novel mechanism of cortical neuroprotection from excitotoxicity-induced necrotic cell death after GDNF application. However, in this study GDNF failed to rescue cortical neurons from apoptotic cell death, thus indicating that GDNF may result in neuroprotection via inhibiting neuronal necrosis, but not apoptosis. Moreover, this study illustrated the indispensable nature of the MAPK (MEK) pathway, and GDNF’s reduction of N-methyl-D-aspartate (NMDA)-triggered calcium influx, resulting in the attenuation of necrotic cell death. However, glutamatergic excitotoxicity induced by non-NMDA agonists (AMPA, α-amino-3-hydroxy-5-methyl-4-isoxazolepropionic acid; kainate) was unable to be attenuated by GDNF administration [26]. Additionally, this study highlighted the fact that GDNF’s neuroprotective effects were likely caused by diminished NMDA receptor activity, and not the result of free radical scavenging. Cheng et al. [51] investigated the downstream neuroprotection signaling of GDNF, and determined that GDNF activated the MAPK signaling pathway, and resulted in increased levels of Bcl-2. Liu et al. [21] described a similar upregulation of Bcl-2 and downregulation of Bax, which provided neuroprotection in vitro and Schwann cell survival in vivo, in rats treated with Schwann cells overexpressing GDNF, as compared to SCI rats. All of these studies provide potential target points within the GDNF signaling pathways for intervention following SCI.

## 6. Studies Employing GDNF for Repair of Experimental SCI

After avulsion injury, axotomized motoneuron cell death was reduced by 50% and somatic atrophy was reduced after treatment with GDNF [52]. In another study of avulsion injury, GDNF administration via AAV-viral vector significantly attenuated spinal cord ventral horn motor neuron death [53]. In one of the earliest studies of GDNF administration after SCI, Ramer et al. [54] reported the ability of GDNF to rescue spinal cord motoneurons. In a contusive SCI model, GDNF showed significant improvement in motor function (Basso Beattie and Bresnahan, (BBB) locomotor rating scale), increased cell survival, and number of spared neuronal fibers compared to phosphate buffered saline (PBS)-controls [51]. Rescuing spinal cord motor neurons from cell death is an important component of sparing motor function after SCI.

Mills et al. [55] described the way in which GDNF enhancement of axonal regeneration occurs within a narrow therapeutic dosage range. Dosage and timing considerations are quite crucial for treatments following SCI. What is an effective dosage at one time point, might be ineffective at a later time point, considering the complex and dynamic lesion environment resulting from SCI. In a compressive clip model of SCI, Kao et al. [56] demonstrated significantly improved motor functional recovery (inclined plane) and reduced infarct zone, in addition to a dramatic increase in the number of VEGF (Vascular endothelial growth factor)-positive and GDNF-positive cells (undetectable in sham and SCI-only groups), and significantly reduced TUNEL (Terminal deoxynucleotidyl transferase (TdT) dUTP Nick-End Labeling) staining. This study suggests the beneficial therapeutic potential of GDNF and VEGF following SCI.

## 7. Studies Using GDNF in Combinational Therapies for Experimental SCI Repair

Iannotti et al. [57] showed robust remyelination, axonal regeneration, and reduced cavitation, as well as modest yet significantly reduced astrogliosis and immune infiltration, in response to GDNF-releasing matrigel guidance channels transplanted following experimental hemisection SCI. Additionally, there was synergistic promotion of axonal regeneration and myelination in response to guidance channels containing both Schwann cells (SCs) and GDNF [57]. This study supports the notion that combinational therapies hold the greatest therapeutic potential for SCI. However, despite significant axonal regrowth into the SCI lesion site, accompanied by the recruitment of myelinating Schwann cells, Blesch and Tuszynski [58] highlighted the difficulty of promoting axonal regrowth through and beyond the lesion site, following secretion of GDNF from genetically modified transplanted fibroblasts. Thus, axonal growth through and beyond the lesion is a major impediment for a majority of SCI therapies.

In a novel study of chronic SCI, using a peripheral nerve graft, GDNF treatment enhanced axonal regeneration 7-fold when compared to controls [59]. In a study with Schwann cell seeded guidance channels, significant axonal regeneration and myelination were observed, along with an increased number of blood vessels within the regenerated tissue, and increased diameter of the regenerated axons [60]. The observed inhibitory astrogliosis was positively modulated, and an intermingling of host and graft tissue was observed at the hemisection lesion interface, in a combinational study of GDNF and Schwann cells in semi-permeable guidance channels [10]. Noteworthy is a study by Zhao et al. [61] in which GDNF reduced axotomy-induced astrogliosis of the facial nerve. In a more recent study, a growth-promoting bridge was formed by the transplantation of Schwann cell seeded guidance channels, with Schwann cells overexpressing GDNF [62]. This GDNF overexpression modulated the astrocytic glial scar, created a more permissive environment for propriospinal axonal regrowth through and beyond the distal end of the lesion, conducted electrical signals through the lesion gap, and improved functional recovery [62]. This study highlights the importance of combinational treatment approaches for traumatic SCI.

In another combinational treatment approach, GDNF was embedded into an alginate hydrogel for slow release, and transplanted in a hemisection SCI model [9]. In this study, GDNF promoted increased functional recovery, increased numbers of intralesional and perilesional neurites, reduced astrogliosis, and increased intralesional vasculature, as compared to controls. Using PLGA (polylactide-co-glycolic acid) microspheres for slow release, Zhang et al. [40] administered GDNF, Chondroitinase ABC, and a Nogo A antibody following a transection SCI. Lu et al. [46] showed remarkably robust axonal regeneration up to 12 mm in length, in a severe SCI transection model (2 mm of cord removed), in a combined therapeutic approach.This included transplantation of neural stem cells in fibrin matrices containing a trophic factor cocktail (GDNF, BDNF (brain-derived neurotrophic factor), PDGF-AA (platelet-derived growth factor), NT3 (neurotrophin-3), IGF-1 (insulin-like growth factor 1), EGF (epidermal growth factor), aFGF (acidic fibroblast growth factor), bFGF (basic fibroblast growth factor), HGF (hepatocyte growth factor), and a calpain inhibitor/). Moreover, this tissue graft resulted in: (1) significantly enhanced motor recovery; (2) significantly improved electrical signals across the lesion gap; (3) survival and differentiation of the neural stem cells; (4) an intermingling of host axons into tissue grafts; (5) increased myelination; and (6) functional synapse formation likely leading to the observed significant improvement in locomotion [46]. This study highlights the potential of various trophic factor combinations, and the necessity for combined therapeutic approaches.

Chen et al. [63] used a combinational approach consisting of hydrogel scaffolds containing Schwann cells which overexpressed GDNF, and were transplanted into the transected rat spinal cord. The observations included increased axonal growth and axon myelination (by host Schwann cells). This study and others demonstrate the potential of Schwann cells for SCI repair. Shahrezaie et al. [64] observed significant functional recovery (BBB) with a combined treatment of bone marrow mesenchymal stem cells (BMSCs) with lentivirus for GDNF expression, more so than SCI alone, BMSCs alone, or BMSCs with an empty lentiviral vector. Another novel combinational treatment approach was utilized by Zhao et al. [61], with a temperature-sensitive heparin-poloxamer hydrogel with high GDNF-binding affinity, orthotopically injected following thoracic compression injury. Rats receiving hydrogel with GDNF showed dramatically increased functional recovery (BBB and inclined plane) compared with hydrogel treatment or SCI alone. Furthermore, this treatment showed reduced astrogliosis, increased axon regeneration, and both autophagy-dependent and autophagy-independent neuroprotection. While many treatments result in neuroprotection, improved functional recovery is more difficult to achieve in models of SCI. In a 2016 study [50], human umbilical cord blood mononuclear cells (hUCB-MCs) were combined with an adenoviral vector containing GDNF, following rat thoracic contusive SCI. Adenoviral-GDNF and hUCB-MCs with adenoviral-GDNF showed significantly more tissue sparing than either of the control groups lacking GDNF. The combined hUCB-MCs with GDNF (adenoviral vector) showed significantly increased myelination compared to hUCB-MCs or adenoviral GDNF alone. Significant functional recovery (BBB) was observed for the adenoviral-GDNF group compared to the adenoviral control; in addition, hUCB-MCs adenoviral-GDNF showed similar improvements to the adenoviral-GDNF group. The GDNF-containing treatment groups also showed distinct changes in various glial cells (astrocytes, oligodendrocytes, and Schwann cells) throughout the injured area, demonstrating other potential targets for SCI repair.

Jiao et al. [65] employed a silk fibroin/alginate GDNF scaffold seeded with human umbilical cord mesenchymal stem cells (hUCMSCs) for a thoracic contusion injury in a rat model. The silk fibroin scaffold combined with alginate had a prolonged release of GDNF when compared to either scaffold alone. Moreover, the combination scaffold, including GDNF seeded with hUCMSCs, resulted in significant functional improvement (BBB), neuroprotection, increased expression of neuronal markers, and significantly reduced inflammatory cytokine expression, compared to: (1) the combination scaffold with GDNF alone; (2) the combination scaffold without GDNF; and (3) SCI alone. A similar combinational study utilized placental-derived mesenchymal stem cells (PMSCs) plus GDNF, compared to bone marrow derived mesenchymal stem cells (BMSCs) plus GDNF accompanied by copolymer scaffolds [66]. Interestingly, PMSCs expressing GDNF did not significantly differ in their SCI repair capability from that of BMSCs expressing GDNF. However, untransfected PMSCs and BMSCs showed significantly less tissue repair than transfected PMSCs and BMSCs expressing GDNF, thus underscoring the high therapeutic potential of GDNF for SCI repair.

In summary, these collective studies demonstrate the beneficial effects of GDNF on multiple cells types within the nervous system, particularly in combinational treatment approaches, for repair of the injured spinal cord. The complex and dynamic milieu resulting from SCI appears to demand combinational treatment approaches for repair and regeneration. Future potential clinical trials might include transfecting the patient’s own Schwann cells to overexpress GDNF, before spinal cord transplantation of the allogenic Schwann cells. This seems like the logical next step for the Schwann cell clinical trial being conducted at the Miami Project to Cure Paralysis [36].

## Figures and Tables

**Figure 1 brainsci-08-00109-f001:**
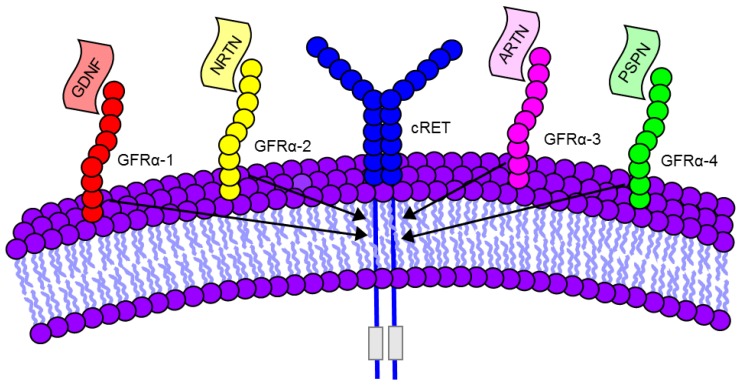
GDNF family of ligands and receptors. GDNF, Neurturin (NRTN), Artemin (ARTN), and Persephin (PSPN). GFRα 1-4 bind to cRET co-receptors (See Table 1).

**Table 1 brainsci-08-00109-t001:** GDNF ligands, receptors, and co-receptors.

Ligand	Receptor	Co-Receptor
GDNF	GFRα 1	cRET
NRTN	GFRα 2	cRET
ARTN	GFRα 3	cRET
PSPN	GFRα 4	cRET

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
