# Peer review of "History of Glial Cell Line-Derived Neurotrophic Factor (GDNF) and Its Use for Spinal Cord Injury Repair"

_brainsci, 2018, doi:10.3390/brainsci8060109_

Round 1

Reviewer 1 Report

Authors reviewed history od GDNF and its effect for  SCI repair.   It is understandble and helpful for readers.

Author Response

We thank the Reviewer for taking time to review our manuscript and graciously provide feedback. We greatly appreciate the comments.

Reviewer 2 Report

Review

Role of GDNF in Spinal Cord Injury Repair – is an important and timely review, specially because there are no effective therapeutic options for spinal cord injury till date.  The manuscript is well written and adequately referenced. 
Indeed, GDNF may serve as an important therapeutic candidate for SCI and all the more so in combination with other agents as suggested by the authors.  Following are two minor suggestions:

1.      Current review is all about experimental spinal cord injury, hence the term experiment SCI should be introduced and emphasized.

2.      A graphical abstract should be included.

Author Response

We thank the Reviewer for taking time to review our manuscript and graciously give us feedback. We appreciate the feedback, and have emphasized ‘experimental’ SCI in our manuscript as well as added the following graphical abstract.

Reviewer 3 Report

The review article by Walker and Xu enumerate the role of GDNF in the SCI repair. There are only a few studies on this interesting topic and surely requires a review at present time. While the author has tried their best to cover all the aspects of GDNF and its role in SCI repair but there is a need to addressed several issues before publication. Some specific examples are as follows: 

Overall, there is a lack of flow. Authors need to present the information in such a way that attracts the audience.

There is a need to correct the pattern of writing the citation in the text, for example, page 1, line 24, Singh, Fehlings et al. Check your manuscript thoroughly for such mistakes.  

Throughout the manuscript, information provided as a conclusion for each study that makes the manuscript non-interested. Authors need to provide more information and their drawbacks and advantages. A tabulated form of all GDNFand SCI studies recommended.     

Authors recently published a review article on VEGF and used some part in the introduction of this manuscript from the review for SCI introduction but fail to mention the citation for the same.

Authors should concise the SCI background, GDNF receptor and localization part. A tabulated form of presentation will be great for GDNF ligand and receptors. 

Latest references should be included.

Authors should conclude the role of GDNF in SCI and its future perspective. 

Instead of so many animals studies, there is lack of clinical status of GDNF and SCI.    

Author Response

We thank the Reviewer for taking time to review our manuscript and provide us with feedback. We appreciate the feedback, and have emphasized experimental SCI in our manuscript, due to the current lack of GDNF clinicial trials. We have modified the title and some details to make it clear that we are providing a background of the GDNF molecule and its role in repair of experimental spinal cord injury.

The citations have been double checked and corrected. A recent review of the literature for the past three years was conducted. We have expanded upon our conclusion and proposed future directions, including clinical applications of GDNF for SCI repair. We have corrected the similar SCI background from our VEGF article introduction. We did find regions of the article which did not flow well, and have corrected those, along with difficult language that impeded flow. We have added conclusions about various studies cited within the article to make it more interesting to the reader, more relevant to the field of SCI, and to draw more conclusions from the referenced studies. We have included a table of GDNF ligands and receptors, as suggested. Again, we appreciate all of the helpful feedback from the Reviewer.